# Gender Differences in the Relationships among Metabolic Syndrome and Various Obesity-Related Indices with Nonalcoholic Fatty Liver Disease in a Taiwanese Population

**DOI:** 10.3390/ijerph18030857

**Published:** 2021-01-20

**Authors:** I-Ting Lin, Mei-Yueh Lee, Chih-Wen Wang, Da-Wei Wu, Szu-Chia Chen

**Affiliations:** 1Division of Endocrinology and Metabolism, Department of Internal Medicine, Kaohsiung Medical University Hospital, Kaohsiung Medical University, Kaohsiung 807, Taiwan; alex780421@gmail.com (I-T.L.); lovellelee@hotmail.com (M.-Y.L.); 2Department of Internal Medicine, Kaohsiung Municipal Siaogang Hospital, Kaohsiung Medical University, Kaohsiung 812, Taiwan; chinwin.wang@gmail.com (C.-W.W.); u8900030@yahoo.com.tw (D.-W.W.); 3Division of Hepatobiliary, Department of Internal Medicine, Kaohsiung Medical University Hospital, Kaohsiung Medical University, Kaohsiung 807, Taiwan; 4Division of Pulmonary and Critical Care Medicine, Department of Internal Medicine, Kaohsiung Medical University Hospital, Kaohsiung Medical University, Kaohsiung 807, Taiwan; 5Division of Nephrology, Department of Internal Medicine, Kaohsiung Medical University Hospital, Kaohsiung Medical University, Kaohsiung 807, Taiwan; 6Faculty of Medicine, College of Medicine, Kaohsiung Medical University, Kaohsiung 807, Taiwan; 7Research Center for Environmental Medicine, Kaohsiung Medical University, Kaohsiung 807, Taiwan

**Keywords:** *non-alcoholic fatty liver disease*, metabolic syndrome, obesity related indices, gender difference

## Abstract

The incidence of *nonalcoholic fatty liver disease*
*(NAFLD)* is increasing worldwide, and it is strongly associated with metabolic syndrome (MetS) and some obesity-related indices. However, few studies have investigated gender differences in these associations. The aim of this study was to investigate associations among MetS and various obesity-related indices with NAFLD, and also look at gender differences in these associations. We enrolled participants who completed a health survey in southern Taiwan. MetS was defined according to the Adult Treatment Panel III for Asians, and the following obesity-related indices were calculated: body mass index (BMI), waist-to-height ratio (WHtR), waist–hip ratio (WHR), lipid accumulation product (LAP), body roundness index (BRI), conicity index (CI), visceral adiposity index (VAI), body adiposity index (BAI), abdominal volume index (AVI), triglyceride-glucose (TyG) index, and hepatic steatosis index (HSI). NAFLD was diagnosed when hepatic steatosis was noted on a liver ultrasound. A total of 1969 (764 men and 1205 women) participants were enrolled. Multivariable analysis showed that both male and female participants with MetS, high BMI, high WHtR, high WHR, high LAP, high BRI, high CI, high VAI, high BAI, high AVI, high TyG index, and high HSI were significantly associated with NAFLD. In addition, the interactions between MetS and gender, WHR and gender, LAP and gender, and TyG index and gender on NAFLD were statistically significant. Among these obesity-related indices, HSI and LAP had the greatest area under the curve in both men and women. Furthermore, stepwise increases in the number of MetS components and the values of indices corresponding to the severity of NAFLD were noted. In conclusion, our results demonstrated significant relationships between MetS and obesity-related indices with NAFLD, and also stepwise increases in the number of MetS components and the values of indices with the severity of NAFLD. MetS, WHR, LAP, and TyG index were associated with NAFLD more obviously in women than in men.

## 1. Introduction

*Nonalcoholic fatty liver disease (NAFLD)* is the most common liver disorder in industrialized Western countries, with a reported prevalence of 6–35% (median 20%) worldwide [1]. Nonalcoholic steatohepatitis (NASH) is a type of NAFLD that is associated with the development of chronic liver disease including cirrhosis and hepatocellular carcinoma. Mantovani et al. demonstrated that NAFLD is associated with a 2.2-fold increased risk of incident diabetes [2]. Compared with patients without NAFLD, patients with NAFLD have a higher risk of fatal and/or nonfatal cardiovascular disease events [3]. The incidence of *NAFLD* is increasing worldwide, and it is strongly associated with metabolic syndrome (MetS). MetS is defined by the World Health Organization as a pathologic condition characterized by abdominal obesity, insulin resistance (IR), hypertension, and hyperlipidemia. Approximately one-third of adults in the United States have been reported to have MetS [4], compared to approximately 16% in Taiwan [5].

Anthropometric indices including lipid accumulation product (LAP), body roundness index (BRI), visceral adiposity index (VAI), abdominal volume index (AVI), conicity index (CI), and body adiposity index (BAI) have been used as surrogate markers of IR and central obesity. These indices have been strongly associated with the diagnosis of MetS and the risk of subsequently developing diabetes or atherosclerotic cardiovascular diseases [6,7,8]. All of these indices are easily calculated and can be quantified using factors including body mass index (BMI), body height (BH), hip circumference (HC), waist circumference (WC), body weight (BW), triglycerides (TGs), and high-density lipoprotein (HDL) cholesterol [6]. In addition, several obesity-related indices including waist to height ratio (WHtR), BMI and WC, and hepatic steatosis index (HSI) have been reported to be effective predictors for the risk of NAFLD [9,10,11]. However, no previous studies have investigated gender differences in the relationships between MetS and various obesity-related indices with NAFLD.

In this study, we used data from more than 2000 people living in southern Taiwan who completed health surveys and investigated associations among MetS and various obesity-related indices with NAFLD. Furthermore, we also explored gender differences in these associations.

## 2. Materials and Methods

### 2.1. Subject Recruitment

We included subjects who took part a health survey from June 2016 to September 2018 in southern Taiwan and were willing to participate in the study. The health survey was promoted through advertisements. All of the included subjects underwent physical examinations and their clinical histories were recorded by an experienced physician. Anthropometric variables (WC, HC, systolic blood pressure (SBP), diastolic blood pressure (DPB), body weight (BW) and body height (BH)) were measured. The exclusion criteria were severe alcohol consumption and a history of hepatitis B and C.

### 2.2. Collection of Demographic, Medical and Laboratory Data

Baseline variables including demographics (age and sex), lifestyle habits (currently smoking and drinking), medical history (hypertension, hyperlipidemia, and diabetes) and SBP and DBP were recorded. The face-to-face interview with researchers included a questionnaire which asked questions on personal information and lifestyle factors. The participants were asked to report the frequency, type, and amount of alcohol they drank. Severe alcohol consumption was defined as heavy drinking is 8 drinks or more per week for women, and 15 drinks or more per week for men. In addition, laboratory data (aspartate aminotransferase (AST), alanine aminotransferase (ALT), fasting glucose, TGs, total, HDL- and low-density lipoprotein (LDL) cholesterol, hemoglobin, uric acid, and estimated glomerular filtration rate (eGFR)) were also recorded at baseline. eGFR was calculated using the Chronic Kidney Disease Epidemiology Collaboration equation (CKD-EPI eGFR) [12]. Hepatitis B antigens and hepatitis C antibodies were assessed to rule out viral hepatitis.

### 2.3. Definition of MetS

MetS was defined according to the National Cholesterol Education Program Adult Treatment Panel (NCEP-ATP) III guidelines [13] and modified criteria for Asians [14] as the presence of three or more of the following five criteria: (1) high blood pressure (SBP ≥ 130 mmHg, DBP ≥ 85 mmHg), or diagnosed hypertension or receiving treatment for hypertension; (2) hyperglycemia (fasting whole-blood glucose concentration ≥110 mg/dL or a diagnosis of diabetes); (3) low HDL cholesterol concentration (<40 mg/dL in men and <50 mg/dL in women); (4) hypertriglyceridemia (TG concentration ≥150 mg/dL); (5) abdominal obesity (WC >90 cm in men and >80 cm in women).

### 2.4. Calculations of Obesity-Related Indices

The obesity-related indices BMI, WHtR, WHR, LAP, BRI, CI, VAI, BAI, AVI, triglyceride-glucose (TyG) index, and HSI were calculated using the following equations:

BMI = BW (kg)/BH^2^ (m);

WHtR = WC (cm)/BH (cm);

WHR = WC (cm)/HC (cm).

LAP was calculated as:

LAP = $\left({\;\mathrm{WC}}_{\left(\mathrm{cm}\right)}-65\right)\times {\;\mathrm{TG}}_{\left(\mathrm{mmol}/L\right)}$ in males;

LAP = $\left({\;\mathrm{WC}}_{\left(\mathrm{cm}\right)}-58\right)\times {\;\mathrm{TG}}_{\left(\mathrm{mmol}/L\right)}$ in females [15].

BRI was calculated as:

BRI = $364.2-365.5\;\times \;\sqrt{1-{\left(\frac{\frac{{\mathrm{WC}}_{\left(m\right)}}{2\pi }}{0.5\;\times {\;\mathrm{BH}}_{\left(m\right)}}\right)}^{2}}$ [16].

CI was calculated using the Valdez equation based on BW, BH and WC as:

CI = $\frac{{\mathrm{WC}}_{\left(m\right)}}{0.109\;\times \;\sqrt{\frac{{\mathrm{BW}}_{\left(\mathrm{kg}\right)}}{{\mathrm{BH}}_{\left(m\right)}}}}$ [17].

VAI score was calculated as described previously [18] using the following sex-specific equations (with TG levels in mmol/L and HDL cholesterol levels in mmol/L):

VAI = $\left(\frac{{\mathrm{WC}}_{\left(\mathrm{cm}\right)}}{39.68\;+\left(1.88\;\times \;\mathrm{BMI}\right)}\right)\times \left(\frac{{\mathrm{TG}}_{\left(\mathrm{mmol}/L\right)}}{1.03}\right)\times \left(\frac{1.31}{{\mathrm{HDL}}_{\left(\mathrm{mmol}/L\right)}}\right)$ in males;

VAI = $\left(\frac{{\mathrm{WC}}_{\left(\mathrm{cm}\right)}}{36.58\;+\left(1.89\;\times \;\mathrm{BMI}\right)}\right)\times \left(\frac{{\mathrm{TG}}_{\left(\mathrm{mmol}/L\right)}}{0.81}\right)\times \left(\frac{1.52}{{\mathrm{HDL}}_{\left(\mathrm{mmol}/L\right)}}\right)$ in females.

BAI was calculated according to the method of Bergman and colleagues as:

BAI = $\frac{{\;\mathrm{Hip}\;\mathrm{circumference}}_{\left(\mathrm{cm}\right)\;}}{{\mathrm{BH}}_{\left(m\right)}{}^{3/2}}\;-18$ [19].

AVI was calculated as AVI = $\frac{2\;\times \;{\left({\;\mathrm{WC}}_{\left(\mathrm{cm}\right)}\;\right)}^{2}+0.7\;\times \;{\left({\;\mathrm{WC}}_{\left(\mathrm{cm}\right)}-{\mathrm{HC}\;}_{\left(\mathrm{cm}\right)}\right)}^{2}}{1000}$ [20].

TyG index = Ln [fasting TG (mg/dL) × fasting plasma glucose (mg/dL)/2] [21].

HSI = 8×(ALT/AST ratio) + BMI (+2, if female; +2, if diabetes mellitus) [11].

### 2.5. Assessment of NAFLD

NAFLD was diagnosed when hepatic steatosis was noted on liver ultrasound that was not related to acute/chronic liver diseases or secondary hepatic fat accumulation, including the use of steatogenic medications and excessive alcohol consumption [22,23]. Trained and experienced radiologists performed all liver ultrasound examinations, and they were blinded to the clinical diagnoses and biochemical test results of the participants. Subjects with at least two of the following abnormal criteria were defined as having hepatic steatosis: diffusely increased liver near-field ultrasound echo (“bright liver”), increased echogenicity in the liver compared to the kidneys, the gradual attenuation of a far-field ultrasound echo, and vascular blurring [23,24]. The presence and severity of hepatic steatosis was scored as follows: 1 = absent; 2 = mild; 3 = moderate; 4 = severe [25].

### 2.6. Ethics Statement

The study protocol was approved by the Institutional Review Board of Kaohsiung Medical University Hospital (number: KMUHIRB-G(II)-20190011). All participants provided informed consent before entering the study.

### 2.7. Statistical Analysis

Data were expressed as percentage, mean ± standard deviation, or median (25th–75th percentile) for triglycerides. Between-group differences were analyzed using the chi-square test for categorical variables and independent t test for continuous variables. Multiple comparisons among the study participants according to the severity of NAFLD were performed using one-way analysis of variance followed by a Bonferroni-adjusted post hoc test. Associations among MetS and the obesity-related indices with NAFLD were identified using multivariable logistic regression analysis. Multiple adjustment included age, AST, ALT, total cholesterol, LDL cholesterol, hemoglobin, eGFR, and uric acid. Receiver operating characteristic (ROC) curves and areas under curves (AUCs) were used to assess the performance and predictive abilities, respectively, of the various obesity-related indices in identifying NAFLD. A *p* value of less than 0.05 was considered to indicate a statistically significant difference. All statistical analyses were performed using SPSS version 19.0 for Windows (SPSS Inc., Chicago, IL, USA).

## 3. Results

The mean age of the 1969 participants (764 males and 1205 females) was 54.9 ± 13.5 years, and the overall prevalence rate of NAFLD was 42.0%. The clinical characteristics of the participants with and without NAFLD and by gender are compared in Table 1. The male participants with NAFLD were younger, had lower HDL cholesterol, and higher SBP, DBP, prevalence rates of hyperlipidemia and MetS, and higher AST, ALT, fasting glucose, TG, uric acid, BMI, WHtR, WHR, LAP, BRI, CI, VAI, BAI, AVI, TyG index, and HSI compared to the male participants without NAFLD. The female participants with NAFLD were older, had lower HDL cholesterol and eGFR, and higher SBP, DBP, prevalence rates of diabetes, hypertension, hyperlipidemia and MetS, and higher AST, ALT, fasting glucose, TG, LDL cholesterol, uric acid, BMI, WHtR, WHR, LAP, BRI, CI, VAI, BAI, AVI, TyG index, and HSI compared to the female participants without NAFLD.

### 3.1. Determinants of NAFLD

Table 2 shows the determinants of NAFLD in the study participants. In the male participants, after adjusting for age, AST, ALT, total cholesterol, hemoglobin, eGFR, and uric acid, those with MetS (*p* < 0.001), high BMI (*p* < 0.001), high WHtR (*p* < 0.001), high WHR (*p* = 0.024), high LAP (*p* < 0.001), high BRI (*p* < 0.001), high CI (*p* < 0.001), high VAI (*p* < 0.001), high BAI (*p* = 0.002), high AVI (*p* < 0.001), high TyG index (*p* < 0.001), and high HIS (*p* < 0.001) were significantly associated with NAFLD. In the female participants, after multivariable adjustments, those with MetS (*p* < 0.001), high BMI (*p* < 0.001), high WHtR (*p* < 0.001), high WHR (*p* < 0.001), high LAP (*p* < 0.001), high BRI (*p* < 0.001), high CI (*p* < 0.001), high VAI (*p* < 0.001), high BAI (*p* < 0.001), high AVI (*p* < 0.001), high TyG index (*p* < 0.001), and high HIS (*p* < 0.001) were significantly associated with NAFLD.

### 3.2. Interactions between Gender and MetS and Obesity-Related Indices on NAFLD

Analysis of the interactions between gender and MetS and obesity-related indices on NAFLD is also shown in Table 2. The interactions between MetS and gender (*p* = 0.019), WHR and gender (*p* = 0.025), LAP and gender (*p* = 0.001), and TyG index and gender (*p* = 0.012) on NAFLD were statistically significant. However, interactions of the other indices and gender did not achieve significance.

We further performed analysis to evaluate the association of MetS with NAFLD using multivariable logistic regression analysis according to different age decades (<45, 45–55, 55–65 and ≥65 years old) in Table 3. MetS is significantly associated with NAFLD in different age decades, both in male and female participants. However, the interactions between MetS and gender on NAFLD were only statistically significant in the age group of 45–55 years old (*p* = 0.047).

### 3.3. ROC Curve Analysis for the Obesity-Related Indices in Identifying NAFLD

Figure 1 demonstrates the ROC analysis and AUCs of 11 obesity-related indices in identifying NAFLD in men (A) and women (B). Among these obesity-related indices in men, HSI had the greatest AUC (AUC = 0.785), followed by LAP (AUC = 0.750), BMI (AUC = 0.718), VAI (AUC = 0.715), AVI (AUC = 0.700), TyG index (AUC = 0.697), WHtR and BRI (AUC = 0.670), BAI (AUC = 0.616), WHR (AUC = 0.615), and CI (AUC = 0.573). In women, HSI had the greatest AUC (AUC = 0.800), followed by LAP (AUC = 0.774), BMI (AUC = 0.753), TyG index (AUC = 0.747), WHtR and BRI (AUC = 0.735), VAI (AUC = 0.732), AVI (AUC = 0.724), WHR (AUC = 0.688), BAI (AUC = 0.675), and CI (AUC = 0.633).

### 3.4. Association between MetS and the Obesity-Related Indices with the Severity of NAFLD

Table 4 shows comorbidity and the values of obesity-related indices according to the severity of NAFLD in the study participants. The values of all obesity-related indices increased with the severity of NAFLD (from absent to severe, ANOVA *p* < 0.001). There were significant trends of stepwise increases in all obesity-related indices.

Figure 2 shows the significant trends of stepwise increases in the number of MetS components corresponding to the severity of NAFLD in the study participants (*p* < 0.001). The more severe the NAFLD score, the higher the number of MetS components (from 0 to 5): absent (30.6%, 28.4%, 21.0%, 13.2%, 5.3%, and 1.6%); mild (8.9%, 23.3%, 25.4%, 24.0%, 12.4%, and 5.9%); moderate (2.6%, 8.6%, 26.2%, 28.0%, 25.1%, and 9.5%) and severe (0%, 2.4%, 9.5%, 52.4%, 23.8%, and 11.9%).

## 4. Discussion

In this study, we demonstrated gender differences in the associations among MetS and various obesity-related indices with NAFLD. We found that MetS, high BMI, high WHtR, high WHR, high LAP, high BRI, high CI, high VAI, high BAI, high AVI, high TyG index, and high HSI were significantly associated with NAFLD. In addition, the interactions between MetS, WHR, LAP, and TyG index and gender on NAFLD were statistically significant.

The first important finding of this study is that MetS was associated with NAFLD, and that a stepwise increase in the number of MetS components corresponded to the severity of NAFLD. Marchesini et al. reported that patients with MetS and NAFLD had a high risk of NASH, and also that MetS was associated with a high risk of severe fibrosis [26]. Previous studies have reported that IR is one of the hallmarks of NAFLD, and that it plays a pivotal role in the pathogenesis of the disease as it is associated with obesity. Moreover, 70%–80% of obese and diabetic patients have been reported to have NAFLD, and IR appears to play a crucial role in the development of hepatic steatosis [27,28]. IR leads to increases in insulin levels, and this combined with increases in lipolysis and/or fat intake promotes the synthesis of hepatic triglycerides [29]. Changes in the production of adipokines such as leptin, resistin, lipocalin 2, *tumor necrosis factor-α*, chemerin, retinol binding protein 4, and *interleukin*-*6* have been reported in obese individuals, and this can lead to the development of obesity-related metabolic diseases. Several anti-inflammatory adipokines including adiponectin, apelin, fibroblast growth factor 21 are also produced by fat tissue. Adipokines exert effects on the pathogenesis and progression of NAFLD through low-grade inflammation, which are closely associated with NAFLD [30,31]. This could explain the association between NAFLD and MetS, as IR has also been associated with MetS and NAFLD.

The second important finding of this study is that various obesity-related indices, including BMI, WHtR, WHR, LAP, BRI, CI, VAI, BAI, AVI, TyG index, and his, were associated with NAFLD, and that the values increased with the severity of NAFLD. A previous study demonstrated that obesity indices including BMI, WC, WHR, WHtR, AVI, and CI showed good discriminatory ability in the diagnosis of MetS [32], and another study reported a significant association between NAFLD and WHtR [33]. Ballestri S et al. showed ultrasonographic fatty liver indicator (US-FLI) can accurately identify the severity of histology and is also correlated with WC, BMI, and IR with various steatogenic liver diseases [34]. In addition, Nelson SM et al. demonstrated that US-FLI may differentiate steatosis from NASH in the average obese population [35]. Praveenraj et al. reported that BMI, WC and WHR had the best predictive ability for NAFLD in a cohort of morbidly obese women [36]. Other studies have reported associations between increased BMI and WC values and both the presence of NAFLD and an increased risk of the progression of liver disease, particularly in older patients [37,38,39]. This may partly be because NAFLD is more closely associated with visceral fat than subcutaneous fat. Visceral adipose tissue has been associated with higher lipolytic rate, increased IR, and greater release of several profibrogenic and proinflammatory mediators than subcutaneous fat, and this may promote the development and progression of NAFLD [40,41,42,43]. In the present study, we found associations among various obesity-related indices and NAFLD. This may be explained by the fact that the presence of obesity with dyslipidemia and IR can contribute to hepatic steatosis.

Another important finding of this study is that among the obesity-related indices, HSI and LAP had the greatest *AUCs* both in men and women. Furthermore, the interactions among MetS, WHR, LAP, and TyG index and gender on NAFLD were statistically significant. Kim YJ et al. [11] evaluated the correlation between HSI and NAFLD in Korea. They found that HSI had an AUC of 0.812. At values of <30.0 or >36.0, HSI ruled out NAFLD with a sensitivity of 93.1%, or detected NAFLD with a specificity of 92.4%, respectively. In a cross-sectional study, Dai et al. enrolled 40,459 subjects aged ≥18 years, and found that LAP was significantly associated with the presence and severity of NAFLD, and that LAP had high diagnostic accuracy to identify NAFLD in the general population [44]. Moreover, they found that LAP was associated with MetS in Chinese children and adolescents, and that it was a better predictive factor than BMI and WHtR to predict MetS [45]. Another study also demonstrated that LAP is a strong and easily obtainable predictor of NAFLD in childhood [46]. LAP is calculated using one biochemical and one anthropometric parameter, and it can theoretically reflect the accumulation of visceral fat [15]. In the current study, we found that the interactions between MetS, WHR and LAP and gender on NAFLD were statistically significant, and that MetS, WHR, and LAP were more obviously associated with NAFLD in the women than in the men. A previous study demonstrated that WHR was the strongest discriminative factor for NAFLD in postmenopausal women [47]. In addition, Hu et al. evaluated the risk factors for NAFLD, and found that a high triglyceride level was the most relevant factor for NAFLD in men, compared to obesity (defined as BMI  ≥  25 kg/m^2^) in women [48]. Another study showed associations between high WHR and IR markers in normal-weight women with normal thyroid function without type 2 diabetes mellitus [49]. In addition, a cross-sectional study that included 200 postmenopausal women showed that LAP was positively correlated with BMI, WHR, TG, the Homeostasis Model Assessment-IR index, and lipid ratio, and that it was a strong and reliable predictor of MetS in postmenopausal women [50]. Two recent review articles suggested that sex differences do exist in the prevalence, risk factors, fibrosis, and clinical outcomes of NAFLD, and sex will probably be considered in future practice guidelines [51,52].

Endogenous estrogens regulate lipid metabolism and inhibit vascular cell growth, inflammation, and plaque progression in premenopausal women [53]. Menopausal women undergo substantial biological and physiological changes including fat redistribution (i.e., more visceral fat accumulation), dyslipidemia and glucose intolerance, which are associated with increased IR, cardiovascular disease and NAFLD [54,55]. Estrogen plays an important role in regulating the development and deposition of adipose tissue in women, and this may promote the deposition of subcutaneous adipose tissue, thereby contributing to greater increases in subcutaneous and total body fat in women compared to men [56]. The accumulation of visceral fat appears to be inhibited before menopause in women, while the accumulation of abdominal fat tends to occur in men. A previous study reported that younger-aged to middle-aged men tended to have a higher risk of developing NAFLD than women of the same age, but that this difference becomes less prominent after the age of 50–60 years [57]. Another study hypothesized that the increase in the prevalence of NAFLD after menopause suggests that estrogen may have a protective effect against NAFLD and its associated risk factors. The National Health and Nutrition Examination *Survey* (NHANES) study also demonstrated a higher prevalence of NAFLD in postmenopausal women than in premenopausal women, and a lower prevalence of NAFLD in postmenopausal women receiving *hormone replacement* therapy compared to postmenopausal women who did not receive *hormone replacement* therapy [53]. The gender differences in NAFLD observed in the current study may therefore be explained by natural female physiological changes which increase the risks of hyperlipidemia, IR, and visceral fat accumulation.

In this article, we found that 11 obesity-related indices, including BMI, WHtR, WHR, LAP, BRI, CI, VAI, BAI, AVI, TyG index, and HSI, were associated with NAFLD both in men and women. Additionally, HSI and LAP had the greatest *AUCs* for NAFLD both in men and women. Furthermore, MetS, WHR, LAP, and TyG index were associated with NAFLD more obviously in the women than in the men. These indicators can be used to clinically assess whether participants are likely to have NAFLD, especially in high risk populations, such as diabetes, MetS, and obesity populations. HSI and LAP have the highest predictive power, whether for men or women. Clinically, we can give priority to using these two indicators for evaluation. For women with MetS, high WHR, high LAP, and high TyG index, we can provide an early intervention and proper education to prevent progression to NAFLD.

There are several limitations to this study. First, casual relationships between MetS, obesity-related indices, and NAFLD could not be defined due to the cross-sectional design of this study. Second, sleep disorders and economic status were not included in our analysis, and these factors may have influenced the development of fatty liver. Third, all of the participants in this study lived in southern Taiwan, and therefore our results may not be generalizable to other populations. Fourth, insulin resistance was one of the hallmarks of NAFLD, and played a pivotal role in the pathogenesis of the disease [58,59]. However, insulin level was not checked in the study. Therefore, we could not evaluate the association between insulin resistance and NAFLD. Fifth, in our study, we use the severity of NAFLD on ultrasound. Ultrasound of the liver seems to be qualitative and insufficient for scoring. In addition, menopause status is lacking in this study. Therefore, we could not survey the estrogen effect. Finally, only subjects who were willing to participate in the study were included, thereby making it more difficult to interpret confidence intervals and standard errors.

In conclusion, our results demonstrated significant relationships among MetS and obesity-related indices with NAFLD, and also stepwise increases in the number of MetS components and the values of the indices with the severity of NAFLD. MetS, WHR, LAP and TyG index were associated with NAFLD more obviously in the women than in the men.

## Figures and Tables

**Figure 1 ijerph-18-00857-f001:**
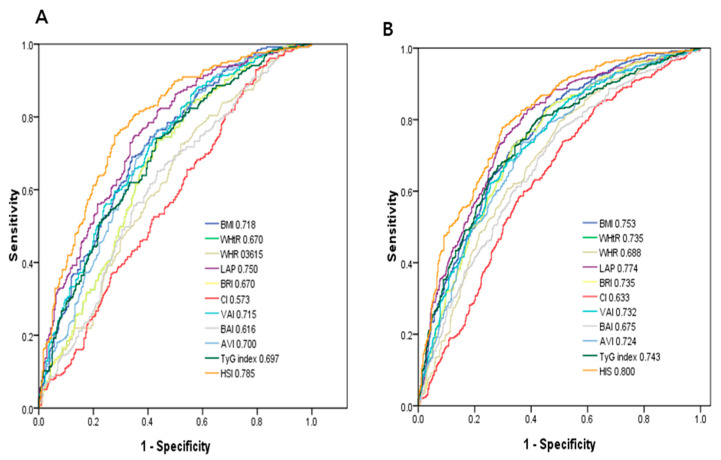
Comparison of the predictive value of 11 obesity-related parameters for diagnosis of NAFLD among (**A**) males and (**B**) females.

**Figure 2 ijerph-18-00857-f002:**
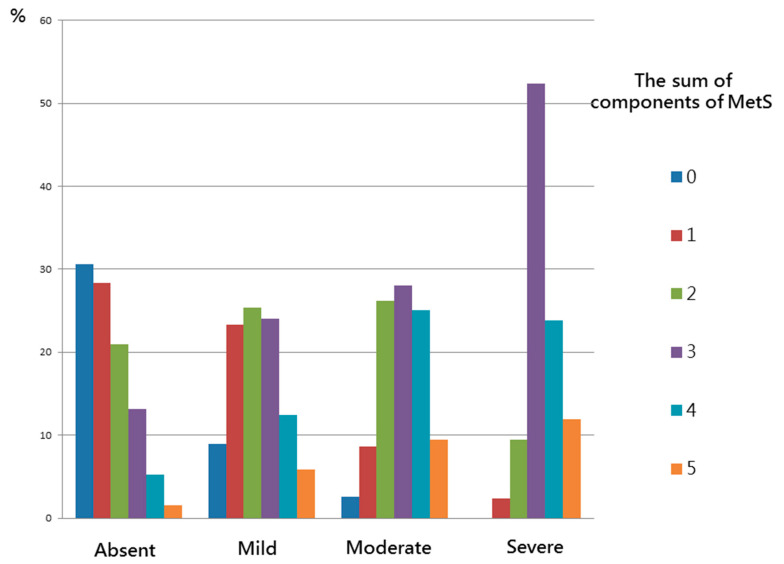
There were significant trends for stepwise increases in the sum of components of MetS corresponding to the severity of NAFLD in study participants.

**Table 1 ijerph-18-00857-t001:** Clinical characteristics of the study participants classified by the presence of different gender and Nonalcoholic fatty liver disease (NAFLD).

Characteristics	Meale (*n* = 764)			Female (*n* = 1205)		
	NAFLD (−) (*n* = 410)	NAFLD (+) (*n* = 354)	*p*	NAFLD (−) (*n* = 733)	NAFLD (+) (*n* = 472)	*p*
The severity of NAFLD			<0.001			<0.001
Absent	100	0		100	0	
Mild	0	50.3		0	54.9	
Moderate	0	40.6		0	40.5	
Sever	0	5.6		0	4.7	
Age (year)	56.36 ± 14.47	53.86 ± 13.41	0.014	53.20 ± 13.85	57.03 ± 11.77	<0.001
Systolic BP (mmHg)	129.83 ± 17.56	133.49 ± 17.13	0.004	128.22 ± 21.25	137.28 ± 19.67	<0.001
Diastolic BP (mmHg)	76.81 ± 10.82	80.73 ± 11.31	<0.001	75.05 ± 12.28	77.74 ± 10.45	<0.001
Current smoking (%)	24.14	28.77	0.148	3.16	1.92	0.194
Diabetes mellitus history (%)	10.00	13.84	0.100	5.32	16.31	<0.001
Hypertension history (%)	26.59	30.23	0.265	17.05	31.78	<0.001
Hyperlipidemia history (%)	2.20	4.80	0.047	0.82	2.75	0.008
MetS (%)	22.98	50.56	<0.001	18.33	55.08	<0.001
Laboratory parameters						
AST (U/L)	26.28 ± 7.93	31.79 ± 14.82	<0.001	24.58 ± 8.05	28.21 ± 13.60	<0.001
ALT (U/L)	23.77 ± 14.56	39.88 ± 28.01	<0.001	18.50 ± 11.50	27.77 ± 17.69	<0.001
Fasting glucose (mg/dL)	98.52 ± 24.16	104.76 ± 30.37	0.002	93.17 ± 18.80	108.39 ± 33.77	<0.001
Triglyceride (mg/dL)	99.5 (74–138.25)	146.5 (100.75–217)	<0.001	82 (61–116.5)	128 (96.25–175)	<0.001
Total cholesterol (mg/dL)	193.04 ± 37.06	196.74 ± 37.49	0.171	203.02 ± 36.94	207.12 ± 37.34	0.062
HDL cholesterol (mg/dL)	49.00 ± 11.08	42.94 ± 8.42	<0.001	60.43 ± 13.71	52.19 ± 11.47	<0.001
LDL cholesterol (mg/dL)	118.56 ± 33.83	121.22 ± 33.06	0.274	117.14 ± 32.26	125.40 ± 36.86	<0.001
eGFR (mL/min/1.73 m^2^)	88.31 ± 13.07	89.86 ± 12.87	0.099	90.51 ± 19.12	87.98 ± 16.88	0.019
Uric acid (mg/dL)	6.40 ± 1.41	6.74 ± 1.57	0.002	4.92 ± 1.25	5.46 ± 1.26	<0.001
Obesity-related indices						
BMI (kg/m^2^)	24.42 ± 3.25	26.97 ± 3.38	<0.001	23.11 ± 3.45	26.64 ± 3.91	<0.001
WHtR	0.51 ± 0.06	0.55 ± 0.05	<0.001	0.49 ± 0.06	0.55 ± 0.06	<0.001
WHR	0.90 ± 0.08	0.92 ± 0.06	0.001	0.82 ± 0.08	0.87± 0.07	<0.001
LAP	29.56 ± 27.01	55.65 ± 44.30	<0.001	21.85 ± 19.51	47.26 ± 43.12	<0.001
BRI	3.67 ± 1.11	4.35 ± 1.11	<0.001	3.31 ± 1.24	4.41 ± 1.27	<0.001
CI	1.23 ± 0.08	1.26 ± 0.07	<0.001	1.18 ± 0.09	1.22 ± 0.09	<0.001
VAI	3.41 ± 2.66	5.84 ± 5.11	<0.001	3.17 ± 2.43	6.16 ± 8.21	<0.001
BAI	26.48 ± 3.53	27.89 ± 3.78	<0.001	30.36 ± 4.37	33.03 ± 4.88	<0.001
AVI	15.08 ± 2.86	17.18 ± 3.15	<0.001	12.38 ± 2.88	14.78 ± 3.19	<0.001
TyG index	8.51 ± 0.56	8.94 ± 0.62	<0.001	8.27 ± 0.53	8.83 ± 0.62	<0.001
HSI	31.7 ± 4.8	36.9 ± 5.2	<0.001	31.1 ± 4.4	36.7 ± 5.2	<0.001

Abbreviations: NAFLD, nonalcoholic fatty liver disease; BP, blood pressure; MetS, metabolic syndrome; AST, aspartate aminotransferase; ALT, alanine aminotransferase; HDL, high-density lipoprotein; LDL, low-density lipoprotein; eGFR, estimated glomerular filtration rate; BMI, body mass index; WHtR, waist-to-height ratio; WHR, waist–hip ratio; LAP, lipid accumulation product; BRI, body roundness index; CI, conicity index; VAI, visceral adiposity index; BAI, body adiposity index; AVI, abdominal volume index; TyG index, triglyceride glucose index; HIS, hepatic steatosis index.

**Table 2 ijerph-18-00857-t002:** Association of MetS and obesity-related indices with fatty liver using multivariable logistic regression analysis.

Characteristics	Male (*n* = 764)			Female (*n* = 1205)			Interaction *p*
	Multivariable			Multivariable			
	OR	95% Confidence Interval	*p*	OR	95% Confidence Interval	*p*	
MetS	2.716	1.914–3.854	<0.001	4.034	2.997–5.428	<0.001	0.019
Obesity-related indices							
BMI (per 1 kg/m^2^)	1.211	1.143–1.282	<0.001	1.257	1.205–1.311	<0.001	0.156
WHtR (per 0.01)	1.120	1.080–1.162	<0.001	1.126	1.099–1.154	<0.001	0.107
WHR (per 0.01)	1.034	1.004–1.065	0.024	1.068	1.044–1.093	<0.001	0.025
LAP (per 1)	1.023	1.016–1.030	<0.001	1.038	1.030–1.045	<0.001	0.001
BRI (per 1)	1.693	1.416–2.024	<0.001	1.766	1.565–1.992	<0.001	0.080
CI (per 0.1)	1.552	1.211–1.990	<0.001	1.401	1.204–1.631	<0.001	0.512
VAI (per 1)	1.212	1.139–1.290	<0.001	1.257	1.191–1.326	<0.001	0.244
BAI (per 1)	1.098	1.034–1.167	0.002	1.114	1.075–1.156	<0.001	0.968
AVI (per 1)	1.220	1.135–1.313	<0.001	1.241	1.176–1.310	<0.001	0.452
TyG index (per 1)	2.888	2.132–3.913	<0.001	4.493	3.387–5.959	<0.001	0.012
HSI (per 1)	1.215	1.153–1.280	<0.001	1.264	1.214–1.315	<0.001	0.707

Values expressed as odds ratio (OR) and 95% confidence interval. Abbreviations are the same as in Table 1. Covariates in the multivariable model included age, AST, ALT, total cholesterol, hemoglobin, eGFR and uric acid.

**Table 3 ijerph-18-00857-t003:** Association of MetS with fatty liver using multivariable logistic regression analysis according to different age decades.

MetS	Male			Female			Interaction *p*
	Multivariable			Multivariable			
	OR	95% Confidence Interval	*p*	OR	95% Confidence Interval	*p*	
Age <45 years old (*n* = 469)	4.637	1.875–11.469	0.001	8.452	3.692–19.352	<0.001	0.340
Age 45–55 years old (*n* = 501)	2.484	1.181–5.224	0.016	5.967	3.086–11.540	<0.001	0.047
Age 55–65 years old (*n* = 496)	3.124	1.431–6.820	0.004	3.763	2.152–6.580	<0.001	0.739
Age ≥65 years old (*n* = 503)	2.645	1.435–4.874	0.002	2.753	1.586–4.778	<0.001	0.707

Values expressed as odds ratio (OR) and 95% confidence interval. Abbreviations are the same as in Table 1. Covariates in the multivariable model included age, AST, ALT, total cholesterol, hemoglobin, eGFR, and uric acid.

**Table 4 ijerph-18-00857-t004:** Comorbidity and the values of obesity-related indices according to the severity of NAFLD in study participants.

	Absent (*n* = 1143)	Mild (*n* = 437)	Moderate (*n* = 347)	Severe (*n* = 42)	*p*
Diabetes mellitus history (%)	7.0	13.0 *	18.4 *	11.9	<0.001
Hypertension history (%)	20.5	29.7 *	33.7 *	23.8	<0.001
Hyperlipidemia history (%)	1.3	3.2	4.0 *	4.8	0.006
Obesity-related indices					
BMI (kg/m^2^)	23.58 ± 3.44	25.66 ± 3.03 *	27.68 ± 3.77 *^,†^	31.06 ± 4.23 *^,†,#^	<0.001
WHtR	0.50 ± 0.06	0.53 ± 0.05 *	0.56 ± 0.06 *^,†^	0.61 ± 0.05 *^,†,#^	<0.001
WHR	0.85 ± 0.09	0.88 ± 0.07 *	0.90 ± 0.07 *^,†^	0.93 ± 0.07 *^,†^	<0.001
LAP	24.62 ± 22.78	42.36 ± 35.44 *	58.58 ± 50.38 *^,†^	75.52 ± 43.83 *^,†,#^	<0.001
BRI	3.44 ± 1.21	4.05 ± 1.03 *	4.66 ± 1.25 *^,†^	5.60 ± 1.18 *^,†,#^	<0.001
CI	1.20 ± 0.09	1.22 ± 0.08 *	1.24 ± 0.08 *^,†^	1.28 ± 0.08 *^,†^	<0.001
VAI	3.26 ± 2.51	5.21 ± 4.79 *	6.96 ± 9.27 *^,†^	6.67 ± 3.95 *	<0.001
BAI	28.95 ± 4.49	30.26 ± 4.57 *	31.63 ± 5.76 *^,†^	33.73 ± 4.03 *^,†^	<0.001
AVI	13.36 ± 3.15	14.88 ± 3.02 *	16.54± 3.37 *^,†^	19.37 ± 3.74 *^,†,#^	<0.001
TyG index	8.35 ± 0.55	8.74 ± 0.61 *	9.03 ± 0.62 *^,†^	9.08 ± 0.54 *^,†^	<0.001
HSI	31.3 ± 4.6	34.9 ± 4.2 *	38.4 ± 5.1 *^,†^	42.9 ± 5.5 *^,†,#^	<0.001

Abbreviations are the same as in Table 1. * *p* < 0.05 compared with non-NAFLD; † *p* < 0.05 compared with mild NAFLD; # *p* < 0.05 compared with moderate NAFLD.

## Data Availability

The data underlying this study is from the Taiwan Biobank. Due to restrictions placed on the data by the Personal Information Protection Act of Taiwan, the minimal data set cannot be made publicly available. Data may be available upon request to interested researchers. Please send data requests to: Szu-Chia Chen, PhD, MD. Division of Nephrology, Department of Internal Medicine, Kaohsiung Medical University Hospital, Kaohsiung Medical University.

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
