# Peer review of "Gender Differences in the Relationships among Metabolic Syndrome and Various Obesity-Related Indices with Nonalcoholic Fatty Liver Disease in a Taiwanese Population"

_ijerph, 2021, doi:10.3390/ijerph18030857_

Round 1
Reviewer 1 Report
The study entitled “Gender differences in the relationships among metabolic syndrome and various obesity-related indices with nonalcoholic fatty liver disease in a Taiwanese population” aims to investigate associations among metabolic syndrome and various obesity-related indices with NAFLD, and gender differences in these associations in a group of participants from the South of Taiwan. The first finding was that metabolic syndrome was associated with NAFLD, and that a stepwise increase in the number of MetS components corresponded to the severity of NAFLD. The second important finding of the study was that various obesity-related indices were associated with NAFLD, and that the values increased with the severity of NAFLD.
The study provides evidence about possible markers to be considered in patients susceptible to NAFLD. Furthermore, gender differences are highlighted since those markers could vary depending on the gender.
After a careful revision I consider that the paper is technically sound. The objective is clearly stated and the introduction provides strong evidence about the background. The experimental data supports the conclusion, and it is well analyzed and clearly presented. The discussion is well written and supported by previous literature.
I would suggest two minor corrections to improve the article before publication:
- Figure 2 must include the title of the axis Y.
- Since insulin resistance is one of the hallmarks of NAFLD, and that it plays a pivotal role in the pathogenesis of the disease as it is associated with obesity, why the authors did not present data of fasting insulin levels or insulin resistance indices?
Author Response
The study entitled “Gender differences in the relationships among metabolic syndrome and various obesity-related indices with nonalcoholic fatty liver disease in a Taiwanese population” aims to investigate associations among metabolic syndrome and various obesity-related indices with NAFLD, and gender differences in these associations in a group of participants from the South of Taiwan. The first finding was that metabolic syndrome was associated with NAFLD, and that a stepwise increase in the number of MetS components corresponded to the severity of NAFLD. The second important finding of the study was that various obesity-related indices were associated with NAFLD, and that the values increased with the severity of NAFLD.
The study provides evidence about possible markers to be considered in patients susceptible to NAFLD. Furthermore, gender differences are highlighted since those markers could vary depending on the gender.
After a careful revision I consider that the paper is technically sound. The objective is clearly stated and the introduction provides strong evidence about the background. The experimental data supports the conclusion, and it is well analyzed and clearly presented. The discussion is well written and supported by previous literature.
I would suggest two minor corrections to improve the article before publication:
- Figure 2 must include the title of the axis Y.
- Ans: Thank you for your suggestion. We have added the title of the axis Y (%) of Figure 2.
- Since insulin resistance is one of the hallmarks of NAFLD, and that it plays a pivotal role in the pathogenesis of the disease as it is associated with obesity, why the authors did not present data of fasting insulin levels or insulin resistance indices?
- Ans: Thank you for your comments. We totally agreed that insulin resistance is one of the hallmarks of NAFLD, and plays a pivotal role in the pathogenesis of the disease. However, this study is for general health survey, fasting insulin level is not checked. Therefore, we do not have insulin resistance data. We have added ths issue in the limitation.
- Fourth, insulin resistance was one of the hallmarks of NAFLD, and played a pivotal role in the pathogenesis of the disease. However, insulin level was not checked in the study. Therefore, we could not evaluate the association between insulin resistance and NAFLD. (Page 10, Line 34-36)

Reviewer 2 Report
GENERAL COMMENT
The Authors performed an interesting study on gender differences in the relationships among metabolic syndrome and various obesity-related indices with NAFLD. Some comments may be raised at improving the quality of the manuscript.
SPECIFIC COMMENTS
- The Authors should better explore the role of age on the association of NAFLD with metabolic parameters according to gender status. The study shows that: females with NAFLD were older than those without it while an opposite pattern was observed in males; “MetS, WHR and LAP were associated with NAFLD more obviously in women than in men.” Do the Authors have data about menopause status? If not the prevalence of NAFLD and its association with metabolic parameters may be shown according to different age decades.
- Literature should be updated and relevant studies in the field discussed e.g:
- Strong relationship between fatty liver and diabetes/metabolic syndrome/insulin resistance and cardiovascular disease.
Mantovani A et al., Gut. 2020 Sep 16:gutjnl-2020-322572. doi: 10.1136/gutjnl-2020-322572.
Ballestri S et al., Adv Ther 2020;37:1910-1932
Targher G, et al., J Hepatol. 2016;65:589-600.
- Known association between qualitative and semiquantitative ultrasonographic fatty liver severity and metabolic parameters:
Ballestri S et al. Metabolism. 2017;72:57-65.
Nelson SM, et al. J Ultrasound Med. 2020;39:749-759.
- Gender-related differences in NAFLD and its determinants:
Lonardo A, et al. Hepatology 2019;70:1457-1469.
Lonardo A, Suzuki A. J Clin Med. 2020;9:1278. doi: 10.3390/jcm9051278.
Author Response
GENERAL COMMENT
The Authors performed an interesting study on gender differences in the relationships among metabolic syndrome and various obesity-related indices with NAFLD. Some comments may be raised at improving the quality of the manuscript.
SPECIFIC COMMENTS
- The Authors should better explore the role of age on the association of NAFLD with metabolic parameters according to gender status. The study shows that: females with NAFLD were older than those without it while an opposite pattern was observed in males; “MetS, WHR and LAP were associated with NAFLD more obviously in women than in men.” Do the Authors have data about menopause status? If not the prevalence of NAFLD and its association with metabolic parameters may be shown according to different age decades.
- Ans: Thank you for your comments. Indeed, manopause status is important for the investigation of gender difference. However, we did not ask the status of manopause in the questionnaire. We have added this issue in the Limitation. Besides, we further performed analysis of the association of MetS with fatty liver using multivariable logistic regression analysis according to different age decades (age < 45, 45-55, 55-65, ≧ 65 years old) in Table 3, and found that the interactions among MetS and gender on NAFLD were statistically significant in the age group of 45-55 years old.
- In addition, menopause status is lacking in this study. Therefore, we could not survey the estrogen effect. (Page 10, Line 34-36)
- We further performed analysis to evaluate the association of MetS with NAFLD using multivariable logistic regression analysis according to different age decades (< 45, 45-55, 55-65 and ≧ 65 years old) in Table 3. MetS is significantly associated with NAFLD in different age decades, whether in male or female participants. However, the interactions between MetS and gender on NAFLD were only statistically significant in the age group of 45-55 years old. (Page 6)
Table 3. Association of MetS with fatty liver using multivariable logistic regression analysis according to different age decades
|
MetS |
Male |
|
Female |
|
||||
|
Multivariable |
|
Multivariable |
||||||
|
|
OR |
95% CI |
p |
|
OR |
95% CI |
p |
Interaction p |
|
Age < 45 years old (n = 469) |
4.637 |
1.875-11.469 |
0.001 |
|
8.452 |
3.692-19.352 |
< 0.001 |
0.340 |
|
Age 45-55 years old (n = 501) |
2.484 |
1.181-5.224 |
0.016 |
|
5.967 |
3.086-11.540 |
< 0.001 |
0.047 |
|
Age 55-65 years old (n = 496) |
3.124 |
1.431-6.820 |
0.004 |
|
3.763 |
2.152-6.580 |
< 0.001 |
0.739 |
|
Age ≧ 65 years old (n = 503) |
2.645 |
1.435-4.874 |
0.002 |
|
2.753 |
1.586-4.778 |
< 0.001 |
0.707 |
Values expressed as odds ratio (OR) and 95% confidence interval (CI). Abbreviations are the same as in Table 1. Covariates in the multivariable model included age, AST, ALT, total cholesterol, hemoglobin, eGFR and uric acid.
- Literature should be updated and relevant studies in the field discussed e.g:
- Strong relationship between fatty liver and diabetes/metabolic syndrome/insulin resistance and cardiovascular disease.
- Mantovani A et al., Gut. 2020 Sep 16:gutjnl-2020-322572. doi: 10.1136/gutjnl-2020-322572.
- Ballestri S et al., Adv Ther 2020;37:1910-1932
- Targher G, et al., J Hepatol. 2016;65:589-600.
- Known association between qualitative and semiquantitative ultrasonographic fatty liver severity and metabolic parameters:
- Ballestri S et al. Metabolism. 2017;72:57-65.
- Nelson SM, et al. J Ultrasound Med. 2020;39:749-759.
- Gender-related differences in NAFLD and its determinants:
- Lonardo A, et al. Hepatology 2019;70:1457-1469.
- Lonardo A, Suzuki A. J Clin Med. 2020;9:1278. doi: 10.3390/jcm9051278.
- Ans: Thank you for your suggestion. We have added theses references and discussed.
- Mantovani et al. demonstrated that NAFLD is associated with a 2.2-fold increased risk of incident diabetes [2]. Compared with patients without NAFLD, patients with NAFLD have a higher risk of fatal and/or non-fatal cardiovascular disease events [3]. (Page 2, Line 12-15)
- Ballestri S et al. showed ultrasonographic fatty liver indicator(US-FLI) can accurately identify the severity of histology and is also correlated with WC, BMI and IR with various steatogenic liver diseases [33]. In addition, Nelson SM et al. demonstrated that US‐FLI may differentiate steatosis from NASH in the average obese population [34]. (Page 9, Line 18-21)
- Two recent review article suggested that sex differences do exist in the prevalence, risk factors, fibrosis, clinical outcomes of NAFLD, and, sex will probably be considered in future practice guidelines [51, 52]. (Page 10, Line 6-8)

Reviewer 3 Report
- The authors use only liver ultrasound to score NAFLD. Ultrasound of the liver seems to be qualitative and insufficient for scoring. It is necessary to discuss the details of scoring as it is an important part of this report.
- This paper cites the involvement of estrogen in the sex differences in the association between metabolic syndrome and various obesity-related indices and nonalcoholic fatty liver disease. They speculate on this point based on previous reports. If their data were available, the authors could show this by comparing premenopausal and postmenopausal women in the present study. They need to mention this point.
Author Response
- The authors use only liver ultrasound to score NAFLD. Ultrasound of the liver seems to be qualitative and insufficient for scoring. It is necessary to discuss the details of scoring as it is an important part of this report.
- Ans: Thank you for your comments. NAFLD has some non-invasive scores, such as fatty liver index, Kotronen score and hepatic steatosis index. However, these indices or scores were made up by markers like BMI, waist circumference, TG, cholesterol, which were X variables in our study. Therefore, theses indices are not suitable as Y variable in our study. We have added the issue in the Limitation.
- Fifth, in our study, we use the severity of NAFLD on ultrasound. ultrasound of the liver seems to be qualitative and insufficient for scoring. However, some non-invasive scores, such as fatty liver index, Kotronen score and hepatic steatosis index, were made up by markers like BMI, WC, TG or cholesterol, which were X variables in the analysis. Therefore, theses indices are not suitable as Y variable in our study. (Page 10, Line 36-41)
- This paper cites the involvement of estrogen in the sex differences in the association between metabolic syndrome and various obesity-related indices and nonalcoholic fatty liver disease. They speculate on this point based on previous reports. If their data were available, the authors could show this by comparing premenopausal and postmenopausal women in the present study. They need to mention this point.
- Ans: Thank you for your comments. Indeed, manopause status is important for the investigation of gender difference. However, we did not ask the status of manopause in the questionnaire. We have added this issue in the Limitation. Besides, we further performed analysis of the association of MetS with fatty liver using multivariable logistic regression analysis according to different age decades (age < 45, 45-55, 55-65, ≧ 65 years old) in Table 3, and found that the interactions among MetS and gender on NAFLD were statistically significant in the age group of 45-55 years old.
- In addition, menopause status is lacking in this study. Therefore, we could not survey the estrogen effect. (Page 10, Line 34-36)
- We further performed analysis to evaluate the association of MetS with NAFLD using multivariable logistic regression analysis according to different age decades (< 45, 45-55, 55-65 and ≧ 65 years old) in Table 3. MetS is significantly associated with NAFLD in different age decades, whether in male or female participants. However, the interactions between MetS and gender on NAFLD were only statistically significant in the age group of 45-55 years old. (Page 6)
Table 3. Association of MetS with fatty liver using multivariable logistic regression analysis according to different age decades
|
MetS |
Male |
|
Female |
|
||||
|
Multivariable |
|
Multivariable |
||||||
|
|
OR |
95% CI |
p |
|
OR |
95% CI |
p |
Interaction p |
|
Age < 45 years old (n = 469) |
4.637 |
1.875-11.469 |
0.001 |
|
8.452 |
3.692-19.352 |
< 0.001 |
0.340 |
|
Age 45-55 years old (n = 501) |
2.484 |
1.181-5.224 |
0.016 |
|
5.967 |
3.086-11.540 |
< 0.001 |
0.047 |
|
Age 55-65 years old (n = 496) |
3.124 |
1.431-6.820 |
0.004 |
|
3.763 |
2.152-6.580 |
< 0.001 |
0.739 |
|
Age ≧ 65 years old (n = 503) |
2.645 |
1.435-4.874 |
0.002 |
|
2.753 |
1.586-4.778 |
< 0.001 |
0.707 |
Values expressed as odds ratio (OR) and 95% confidence interval (CI). Abbreviations are the same as in Table 1. Covariates in the multivariable model included age, AST, ALT, total cholesterol, hemoglobin, eGFR and uric acid.

Reviewer 4 Report
Lin IT et al. reported the associations among the metabolic syndrome (MetS) and various obesity-related indices with non-alcoholic fatty liver disease (NAFLD), and also gender differences in these associations. Multivariable analysis showed that both male and female participants with MetS, high body mass index (BMI), highwaist-to-height ratio
(WHtR), high waist–hip ratio (WHR), high lipid accumulation product (LAP), high body roundness index (BRI), high conicity index (CI), high visceral adiposity index (VAI), high body adiposity index (BAI), high abdominal volume index (AVI) and high triglyceride-glucose index were significantly associated with NAFLD. This reviewer thinks that the approach was quite common and also old, therefore not so much valuable information is included in the draft. The following are my comments.
- Gender differences in the associations among MetS and various obesity-related indices with NAFLD. There are many reports on gender differences.1,2 The authors should specify which points are novel compared to the previous reports.
- Materials and methods. The exclusion criteria were severe alcohol consumption and a history of hepatitis B and C.
→ a) The authors should describe details of what tests were done to rule out viral hepatitis.
- b) The authors should describe in detail how you checked daily alcohol consumption.
- Table 1. The proportion of severity of hepatic steatosis should add in the Table 1.
- Table 2. Multivariate analysis.
- a) The authors should describe in detail about the method of multivariate analysis in “Statistical analysis” section.
- b) The authors should analyze with multi-colinality in mind.
- Table 3. severity of hepatic steatosis. Since it is not quantitative to diagnose the severity of fatty liver based on echo findings alone, the number of MetS should be analyzed by grouping into 4 groups using the interquartile range of the Fatty liver index.3
- The references are biased towards older ones.
References
- Marchesini G, Bugianesi E, Forlani G, et al. Nonalcoholic fatty liver, steatohepatitis, and the metabolic syndrome. Hepatology. 2003;37(4):917-923.
- Hamaguchi M, Takeda N, Kojima T, et al. Identification of individuals with non-alcoholic fatty liver disease by the diagnostic criteria for the metabolic syndrome. World J Gastroenterol. 2012;18(13):1508-1516.
- Bedogni G, Bellentani S, Miglioli L, et al. The Fatty Liver Index: a simple and accurate predictor of hepatic steatosis in the general population. BMC Gastroenterol. 2006;6:33.
Author Response
Lin IT et al. reported the associations among the metabolic syndrome (MetS) and various obesity-related indices with non-alcoholic fatty liver disease (NAFLD), and also gender differences in these associations. Multivariable analysis showed that both male and female participants with MetS, high body mass index (BMI), highwaist-to-height ratio (WHtR), high waist–hip ratio (WHR), high lipid accumulation product (LAP), high body roundness index (BRI), high conicity index (CI), high visceral adiposity index (VAI), high body adiposity index (BAI), high abdominal volume index (AVI) and high triglyceride-glucose index were significantly associated with NAFLD. This reviewer thinks that the approach was quite common and also old, therefore not so much valuable information is included in the draft. The following are my comments.
- Gender differences in the associations among MetS and various obesity-related indices with NAFLD. There are many reports on gender differences.1,2 The authors should specify which points are novel compared to the previous reports.
- Ans: Thank you for your comments. Really, previous studies had report the association between MetS and NAFLD, even gender difference. As the reviewer pointed that reference 2 (World J Gastroenterol. 2012;18(13):1508-1516.) have reported the efficiency of the criterion of MetS to detecting NAFLD in different gender. Hu et al. evaluated the risk factors for NAFLD, and found that a high triglyceride level was the most relevant factor for NAFLD in men, compared to obesity (defined as BMI ≥ 25 kg/m2) in women (BMC gastroenterology 2012, 12, (1), 123). In our study, except MetS, BMI and TG, we further expland the markers to waist-to-height ratio (WHtR), waist–hip ratio (WHR), lipid accumulation product (LAP), body roundness index (BRI), conicity index (CI), visceral adiposity index (VAI), body adiposity index (BAI), abdominal volume index (AVI) and triglyceride-glucose (TyG) index. Besides, we also compare the predictive value of 10 obesity-related parameters for diagnosis of NAFLD among males and females, which is different from previous articles. Therefore, we think, the association between various obesity-related indices and predictive values with NAFLD is the innovation of this article.
- Materials and methods. The exclusion criteria were severe alcohol consumption and a history of hepatitis B and C.
- The authors should describe details of what tests were done to rule out viral hepatitis.
- Ans: We have checked hepatitis B antigen and hepatitis C antibody to rule out viral hepatitis. We have added in the Methods.
- Hepatitis B antigen and hepatitis C antibody were checked to rule out viral hepatitis. (Page 3, Line 10-11)
- The authors should describe in detail how you checked daily alcohol consumption.
- Ans: We have added in the Methods.
- The face-to-face interview with researchers included a questionnaire which asked about personal information and lifestyle factors. The participants were asked to report the frequency, type, and amount of alcohol. Severe alcohol consumption was defined as heavy drinking is 8 drinks or more per week for women, and 15 drinks or more per week for men. (Page 3, Line 3-6)
- Table 1. The proportion of severity of hepatic steatosis should add in the Table
- Ans: We have added the proportion of severity of hepatic steatosis in Table 1.
- Table 2. Multivariate analysis.
- The authors should describe in detail about the method of multivariate analysis in “Statistical analysis” section.* Multiple adjustment included age, AST, ALT, total cholesterol, LDL-cholesterol, hemoglobin, eGFR and uric acid. (Page 4, Line 18-20)Ans: Thank you for your kind remind. We found that there is a high correlation between total cholesterol and LDL-cholesterol (r = 0.884, p < 0.001). Therefore, we delete LDL-cholesterol in the multivariable analysis (Table 2) to minimize multi-colinality.
- b) The authors should analyze with multi-colinality in mind.
- Ans: We have added in “Statistical analysis” section.
- Table 3. severity of hepatic steatosis. Since it is not quantitative to diagnose the severity of fatty liver based on echo findings alone, the number of MetS should be analyzed by grouping into 4 groups using the interquartile range of the Fatty liver index.3
Ans: Thank you for your comments. NAFLD has some non-invasive scores, such as fatty liver index, Kotronen score and hepatic steatosis index. However, these indices or scores were made up by markers like BMI, waist circumference, TG, cholesterol, which were X variables in our study. Therefore, theses indices are not suitable as Y variable in our study. We have added the issue in the Limitation.
- Fifth, in our study, we use the severity of NAFLD on ultrasound. ultrasound of the liver seems to be qualitative and insufficient for scoring. However, some non-invasive scores, such as fatty liver index, Kotronen score and hepatic steatosis index, were made up by markers like BMI, WC, TG or cholesterol, which were X variables in the analysis. Therefore, theses indices are not suitable as Y variable in our study. (Page 10, Line 36-41)
- The references are biased towards older ones.
Ans: Thank you for your comments. We have added newer references (2, 3, 33, 34, 51, 52).
- Mantovani et al. demonstrated that NAFLD is associated with a 2.2-fold increased risk of incident diabetes [2]. Compared with patients without NAFLD, patients with NAFLD have a higher risk of fatal and/or non-fatal cardiovascular disease events [3]. (Page 2, Line 12-15)
- Ballestri S et al. showed ultrasonographic fatty liver indicator(US-FLI) can accurately identify the severity of histology and is also correlated with WC, BMI and IR with various steatogenic liver diseases [33]. In addition, Nelson SM et al. demonstrated that US‐FLI may differentiate steatosis from NASH in the average obese population [34]. (Page 9, Line 18-21)
- Two recent review article suggested that sex differences do exist in the prevalence, risk factors, fibrosis, clinical outcomes of NAFLD, and, sex will probably be considered in future practice guidelines [51, 52]. (Page 10, Line 6-8)

Round 2
Reviewer 2 Report
I feel the manuscript has improved.
This sentence is not clear "Fifth, in our study, we use the severity of NAFLD on ultrasound. ultrasound of the liver seems to be qualitative and insufficient for scoring. However, some non-invasive scores, such as fatty liver index, Kotronen score and hepatic steatosis index, were made up by markers like BMI, WC, TG or cholesterol, which were X variables in the analysis. Therefore, theses indices are not suitable as Y variable in our study. " Please rearrange/clarify.
Author Response
I feel the manuscript has improved.
- This sentence is not clear "Fifth, in our study, we use the severity of NAFLD on ultrasound. Ultrasound of the liver seems to be qualitative and insufficient for scoring. However, some non-invasive scores, such as fatty liver index, Kotronen score and hepatic steatosis index, were made up by markers like BMI, WC, TG or cholesterol, which were X variables in the analysis. Therefore, theses indices are not suitable as Y variable in our study. " Please rearrange/clarify.
Ans: Thank you for your question. Last time, the reviewer suggested using fatty liver index as outcomes (Y variable) instead of severity of fatty liver because ultrasound of the liver seems to be qualitative and insufficient for scoring. However, fatty liver index, Kotronen score and hepatic steatosis index, were made up by markers like BMI, WC, TG or total cholesterol , and obesity-related indices in our study were also composited by these variables. If we use fatty liver index, Kotronen score and hepatic steatosis index as outcomes (Y variable), and obesity-related indices were X variables in the binary logistic regression analysis, and then multi-collinearity would be too high. Therefore, we just added hepatic steatosis index into X variable. We have changed the sentence to “"Fifth, in our study, we use the severity of NAFLD on ultrasound. Ultrasound of the liver seems to be qualitative and insufficient for scoring.”

Reviewer 3 Report
Dr. I-Ting Lin et al. have reported that significant relationships between MetS and obesity-related indices with NAFLD, and also stepwise increases in the number of the metabolic syndrome (MetS) components and the values of indices with the severity of NAFLD. MetS, WHR and LAP were associated with NAFLD more obviously in women than in men. In there proofread version, they showed us about interactions between gender and MetS and obesity-related indices in NAFLD with Table 3.
This information is very interesting and important for the clinician. The authors have proofread everything clearly. I believe that it is good enough to accept this manuscript for publication in International Journal of Environmental Research and Public Health.
Author Response
Dr. I-Ting Lin et al. have reported that significant relationships between MetS and obesity-related indices with NAFLD, and also stepwise increases in the number of the metabolic syndrome (MetS) components and the values of indices with the severity of NAFLD. MetS, WHR and LAP were associated with NAFLD more obviously in women than in men. In there proofread version, they showed us about interactions between gender and MetS and obesity-related indices in NAFLD with Table 3.
This information is very interesting and important for the clinician. The authors have proofread everything clearly. I believe that it is good enough to accept this manuscript for publication in International Journal of Environmental Research and Public Health.
Ans: Thank you for your review to make our article better.

Reviewer 4 Report
Authors:
Besides, we also compare the predictive value of 10 obesity-related parameters for diagnosis of NAFLD among males and females, which is different from previous articles. Therefore, we think, the association between various obesity-related indices and predictive values with NAFLD is the innovation of this article.
→ If 10 parameters are responsible for the sex difference in NAFLD, what is their clinical significance? It's hard to imagine using these 10 parameters in a real clinical situation. In fact, which parameters are the best? The authors should describe about this point in “Discussion section”.
Authors:
Table 4. This reviewer thinks that the authors should present the data comparing Fatty Liver Index by severity of fatty liver.
Table 4. This reviewer also thinks that the authors should present the data comparing Fatty Liver Index by gender.
Table 4. This reviewer also thinks that the authors should present the data comparing history of DM, hypertension, and dyslipidemia by severity of fatty liver.
Authors: p10, Fourth, insulin resistance was one of the hallmarks of NAFLD, and played a pivotal role in the pathogenesis of the disease.
→ The authors should cite the recent articles listed below.1,2
References
- Khan RS, Bril F, Cusi K, Newsome PN. Modulation of Insulin Resistance in Nonalcoholic Fatty Liver Disease. Hepatology. 2019;70(2):711-724.
- Fujii H, Kawada N, Japan Study Group Of Nafld J-N. The Role of Insulin Resistance and Diabetes in Nonalcoholic Fatty Liver Disease. Int J Mol Sci. 2020;21(11).
Author Response
- Besides, we also compare the predictive value of 10 obesity-related parameters for diagnosis of NAFLD among males and females, which is different from previous articles. Therefore, we think, the association between various obesity-related indices and predictive values with NAFLD is the innovation of this article.→ If 10 parameters are responsible for the sex difference in NAFLD, what is their clinical significance? It's hard to imagine using these 10 parameters in a real clinical situation. In fact, which parameters are the best? The authors should describe about this point in “Discussion section”.Ans: Thank you for your comments. We have added the issue in the Discussion.
- In this article, we found that 11 obesity-related indices, including BMI, WHtR, WHR, LAP, BRI, CI, VAI, BAI, AVI, TyG index, and HSI were associated with NAFLD both in men and women. Besides, HSI and LAP had the greatest AUCs for NAFLD both in men and women. Furthermore, MetS, WHR, LAP and TyG index were associated with NAFLD more obviously in the women than in the men. These indicators can be used clinically to assess whether participants are likely to have NAFLD, especially in high risk population, such as diabetes, MetS and obesity. HSI and LAP have the highest predictive power, whether men or women. Clinically, we can give priority to using these two indicators for evaluation. For women with Mets, high WHR, high LAP and high TyG index, we can give early intervention and proper education to prevent progression to NAFLD. (Page 10, Line 46 to Page 11, Line 7)
2. Table 4. This reviewer thinks that the authors should present the data comparing Fatty Liver Index by severity of fatty liver. This reviewer also thinks that the authors should present the data comparing Fatty Liver Index by gender. Ans: Thank you for your suggestion. Fatty liver index requires GGT, however, we don’t have GGT level in the study. We use hepatic steatosis index (HSI) instead of fatty liver index. We have added HSI in Table 2 to compare HSI by gender, in Table 4 to compare HSI by severity of fatty liver, and in Figure 1 to add HIS in the predictive value of 11 obesity-related parameters for diagnosis of NAFLD among (A) males and (B) females.
3. Table 4. This reviewer also thinks that the authors should present the data comparing history of DM, hypertension, and dyslipidemia by severity of fatty liver. Ans: Thank you for your suggestion. We have added the data comparing history of DM, hypertension, and dyslipidemia by severity of fatty liver in Table 4.p10, Fourth, insulin resistance was one of the hallmarks of NAFLD, and played a pivotal role in the pathogenesis of the disease.→ The authors should cite the recent articles listed below.1,2
References
1.Khan RS, Bril F, Cusi K, Newsome PN. Modulation of Insulin Resistance in Nonalcoholic Fatty Liver Disease. Hepatology. 2019;70(2):711-724.
2.Fujii H, Kawada N, Japan Study Group Of Nafld J-N. The Role of Insulin Resistance and Diabetes in Nonalcoholic Fatty Liver Disease. Int J Mol Sci. 2020;21(11).
Ans: Thank you for your suggestion. We have added these 2 references (59 and 60).
